# β-Cyclodextrin-Mediated Beany Flavor Masking and Textural Modification of an Isolated Soy Protein-Based Yuba Film

**DOI:** 10.3390/foods9060818

**Published:** 2020-06-22

**Authors:** Eun-Jung Lee, Honggyun Kim, Jong Yeop Lee, Karna Ramachandraiah, Geun-Pyo Hong

**Affiliations:** Department of Food Science and Biotechnology, Sejong University, Seoul 05006, Korea; eunjunglee@sejong.ac.kr (E.-J.L.); vollry@sejong.ac.kr (H.K.); kmc2238@naver.com (J.Y.L.)

**Keywords:** isolated soy protein, yuba film, β-cyclodextrin, beany flavor, texture

## Abstract

The application of β-cyclodextrin (CD) to remove unattractive volatile compounds has been applied in various food products. This study investigated the effect of CD concentration (1–4%) on the beany flavor masking and textural modification of yuba film prepared by isolated soy protein (ISP) in the presence of (+CD), or after removing, the flavor-entrapped CD (−CD). Based on gas chromatography–mass spectrometry (GC–MS), the addition of CD caused a decrease in 1-octen-3-ol, benzaldehyde, hexanal, and 2-heptanone, which are characterized as the major beany flavor compounds. Regardless of presence or removal, the use of CD was effective in reducing beany flavor in yuba film. Scanning electron microscopy (SEM) observation indicated that the CD present in yuba film was distributed on the lower surface and matrices of the films. In yuba film containing 4% CD, the CD crystals were concentrated on both the upper and lower surfaces of the film. The textural properties of the yuba film were affected by the presence or removal of CD, and better puncture strength was obtained when yuba was made after removing the CD. Therefore, this study indicates that the addition of CD was a good approach to mask the beany flavor of soy protein-based products, and textural properties could be improved by removing CD from the product formulation.

## 1. Introduction

The demand for meat analogs has been increasing due to meat consumption-related concerns, including human health and environmental and ethical concerns. As a result, various texturized plant proteins are being developed for the substitution of animal protein in the human diet [1]. The typical structures fabricated by the texturization of plant proteins are fibers, chunks, slices, granules and films. Among these structures, yuba, a soy protein–lipid film, has gained attention due to its cost effectiveness and environmental friendliness [2]. Yuba film formed on the surface of soymilk, heated on a shallow pan, has the potential to be modified as a meat substitute owing to its flexible, thin characteristics and longitudinal patterns [3,4]. Isolated soy protein (ISP) is a useful material for the development of meat substitutes due to its abundant supply, functional properties, and nutritious values [5]. The merits of the utilization of ISP as a base material for yuba film include product consistency, yield control and the quality of the films. However, a major issue with ISP produced by isoelectric precipitation is the beany flavor originating from soybean [6]. Moreover, for improved consumer acceptance of ISP-based yuba films as a meat substitute, further development in the flavor, appearance, and texture of films is required. 

β-Cyclodextrin (CD), a cyclic oligosaccharide consisting of glucopyranose units, has a hydrophilic outer surface and hydrophobic cavity, which form inclusion complexes with hydrophobic molecules of suitable dimensions [7]. CD is being applied as a nanocarrier owing to its supramolecular structure that resembles a cage [8]. CD, which is formulated by the hydrolytic process of starch degradation, has been considered GRAS since 1998. Moreover, it has been investigated in the improvement of flavor and taste using its inclusion complex-forming ability with various compounds [9,10]. Recently, inclusion complexes with volatile components in the CD cavity have been used for the removal or masking of undesirable compound flavors [11,12]. The limited water solubility of CD, which is lower than that of linear dextrin, is due to its rigid structure caused by hydrogen bonding between the hydroxyl groups and the excess CD that exists as crystals in aqueous suspension [13]. 

CD is a potential ingredient for the improvement of flavor and physical properties of yuba films [14]. In addition, the crystallinity of CD is likely to improve the textural properties of ISP-based products. However, studies on the application of CD for the reduction of beany flavor and the improvement of the textural property of yuba film are limited. Although the application of CD is an effective way of removing the beany flavor from soy formulations, it is unclear whether flavor-entrapped CD is removed or present in soy products. Therefore, this study investigated the effect of CD concentration on the flavor profiles and textural characteristics of yuba film in the presence of (+CD), or after removing, flavor-entrapped CD (−CD). 

## 2. Materials and Methods 

### 2.1. Materials

Food-grade ISP (ISP-TX2001, Shandong Yuxin Bio-Tech Co., Shandong, China) powder and CD (β-cyclodextrin, Mengzhou Hongji Biological Co., Henan, China) were purchased from a local supplier. All chemicals used in this study were of analytical grade.

### 2.2. Preparation of a CD-Treated ISP Suspension

ISP (5%, *w*/*w*) and CD (0, 1, 2, and 4%, *w*/*w*) were suspended in distilled water. The suspension was heated to 90 °C and incubated in a water bath at 90 °C for 15 min. The heated suspension was then cooled to 10 °C with ice slurry and divided into two groups. One group was centrifuged at 3000× *g* for 10 min to remove the excessive flavor-entrapped CD, and the supernatant was used as the −CD treatment. The other group was directly used as a control group (+CD). Numbers 0–4 represent the concentrations of CD (%, *w*/*w*) in ISP colloids. A small sample (~20 g) was taken and used for gas chromatography–mass spectrometry (GC–MS) analysis.

### 2.3. Solid-phase Microextraction Gas Chromatography–mass Spectrometry (SPME GC–MS)

Samples (4 g) were added to 20-mL screw glass vials, and headspace SPME was performed using an autosampler (TriPlus RSH Autosampler, Thermo Scientific, Waltham, MA, USA). Internal standard 1,2-dichlorobenzene-d4 (1 μL/g of sample) was added. An SPME fiber (65 μm polydimethylsiloxane/divinylbenzene (PDMS/DVB) 23 Ga, Supelco Co., Bellefonte, PA, USA) was used for extraction. Vials were incubated for 10 min PDMS/DVB at 60 °C in an autosampler with agitation. Volatile compounds were extracted by placing SPME fiber into the vial and exposing it to the headspace for 30 min at 60 °C [15].

A gas chromatography system (TRACE 1310, Thermo Scientific) attached to triple quadrupole MS (Thermo Scientific) was used. The injection port used was in split mode with a flow rate of 20 mL/min and a surge pressure of 400 kPa. Helium, as the carrier gas, was employed at a constant flow rate of 2 mL/min. A DB-WAX capillary column (60 m × 0.25 mm, film thickness 0.5 μm, Agilent Technologies, Santa Clara, CA, USA) was used for volatile compound separation. The initial oven temperature was held at 40 °C for 2 min. It was then raised at 4 °C/min to 150 °C and finally at 10 °C/min to 240 °C, where the temperature remained constant for 5 min. The volatile compounds were identified by comparing the results with mass spectra and retention indexfrom a database developed by NIST library guidelines. The values were quantified by calculating the ratio of the peak area of each volatile compound to the peak area of the internal standard.

### 2.4. Preparation of Yuba Film

The ISP suspension with and without CD was poured into a rectangular pan (12.9 × 17.8 × 2.3 cm) and heated at 85 °C using a thermostat water bath (SL-WB8D, Sunil Science, Jinju, Korea). The films formed on the mixture surface were picked up with glass rods at 25-min intervals. Every 3rd film formed was chosen for physical analysis. Skimmed films were hung at room temperature for 20 min before being spread in plastic bags to form sheets.

### 2.5. Physical Analysis

Physical properties of ISP-based yuba film were evaluated with a color and puncture test. The color of films folded 3 times were measured by a color meter (CM-5, Minolta, Japan) utilizing the CIELAB *L^*^a^*^b^*^* scale. The values of *L^*^*, *a^*^* and *b^*^* were recorded. A puncture test was performed to measure the strength of the single-layer film. To control the moisture content of the films during the puncture test, films were placed in a closed chamber along with wet paper overnight at 4 °C. Mechanical properties were determined using a texture analyzer (CT3, Brookfield, Middleboro, MA, USA) using the modified method of Gontard et al. [16]. Tests for each film were performed in triplicate, with films being placed on a puncture mold (14 mm) beneath a cylindrical probe that has a smooth edge with a 12.7-mm diameter (TA5, Brookfield). Films were punctured at a speed of 1 mm/s with the probe moving perpendicular to the film. Puncture deformation was expressed as the change in length from the original position to the rupture point of the film, and the puncture strength was calculated as the maximum force applied to the film before a puncture was achieved.

### 2.6. Sensory Analysis 

The flavor strength, beany flavor, and sweet flavor of ISP-based yuba films were evaluated using a 9-point intensity rating (0 extremely weak to 9 extremely strong). Ten trained panelists from Sejong University were involved. Each panelist had previous experience in the sensory evaluation of soy products and received training on beany flavor. Samples taken out of the refrigerator were kept at room temperature for 1 h before being dispensed into capped glass vials coded with 3 random digits. The +CD-0 was provided at first to panelists as the standard with score 5, and then randomly served the other samples with offering a 3 min rest between individual samples.

### 2.7. Scanning Electron Microscopy (SEM) Observation

The microstructures of the samples were observed with a tabletop scanning electron microscope (TM4000plus-MA, Hitachi Science System Ltd., Hitachinaka, Japan) at an intensity of 15 kV. 

### 2.8. Statistical Analysis

Statistical analyses were performed using the R program (RStudio 1.1.456, RStudio Inc., Boston, MA, USA). The differences in the sensory evaluation, color parameters, puncture strength, and deformation were analyzed using one-way analysis of variance (ANOVA), and Duncan’s multiple range test was used as a post hoc procedure. Principal component analysis (PCA) was performed using the mean values of relative volatile compounds. PCA was applied to assess the relationships of the volatile compounds among various concentrations of CD and in ISP colloid post processing.

## 3. Results and Discussion

### 3.1. Composition of Volatile Compounds

The experimental data of flavor release from ISP colloids with varying CD concentrations in the +CD and –CD groups are presented in Table 1. Using GC–MS, 36 volatile components were identified as abundant compounds in the samples. As CD concentrations increased (0, 1, 2, and 4%), equivalent headspace concentrations of volatile compounds in +CD groups were also increased due to the larger increases in total alkanes and alkenes originating from CD, while alcohols, acids, ketones, and aldehydes were decreased. In the −CD group, which was devoid of flavor-entrapped CD, equivalent headspace concentrations of volatile compounds decreased with increasing CD concentrations. The total alkanes and alkenes of the −CD group showed restricted increases, and concentration-dependent increases were not observed. With respect to alkanes, the +CD group showed notably increasing trends with increasing concentrations of CD. The most abundant compounds included decan, dodecane, tridecane, hexadecane, 2,4-dimethylheptane, undecane, and 4-methyldecane. However, in the −CD group, the levels of these compounds were similar regardless of the CD concentration, indicating a lack of insoluble CD complexes, which were removed by centrifugation. Likewise, concentration-dependent increases in alkene compounds such as 1,2-demethyl-4-ethylbenzene, dl-limonene, and methylbenzene were found to be limited for the −CD group. In the case of 1,3-di-tert-butylbenzene, concentration-related changes were detected, with 1% CD having the highest peak ratio followed by 2 and 4% CD concentrations. This phenomenon might be due to unknown interference effects caused by flavor compounds in the headspace.

Among the detected compounds, the beany-flavor-related compounds were 1-octen-3-ol, benzaldehyde, hexanal, 2-heptanone, and 2-pentylfuran, which were commonly detected in other studies as compounds showing odor characteristics related to beany flavor [17,18,19]. Of the aforementioned compounds, 1-octen-3-ol, benzaldehyde, hexanal, and 2-heptanone were decreased with increasing CD concentrations regardless of postprocess centrifugation. 2-Pentylfuran adsorbed in headspace SPME was notably increased with increasing CD concentrations in the samples without centrifugation. However, there was a notable decrease in the 4% CD addition (CDcf-4) compared with CDcf-0.

In one study, with the addition of CD to soymilk, a decreased beany odor was reported [20,21]. Suratman et al. (2004) showed compounds associated with a beany flavor in soymilk, such as pentanal, hexanal, hexanol, heptanal, benzaldehyde, 1-octen-3-ol, 2-pentylfuran, and nonanal, in equivalent headspace concentrations using Carboxen^®^/polydimethylsiloxane (CAR/PDMS) fiber. These compounds, with the exception of hexanal, were significantly decreased using 0.5% (*w*/*v*) α- or γ-CD, which varied in the number of glucopyranose units. Recently, Xiaodi et al. (2017) studied the effects of β-CD (0.25–1.00% (*w*/*v*)) addition at different heating temperatures on flavor quality during soymilk processing. They reported decreases in the critical beany flavor compounds, such as hexanal, hexanol, and 1-octen-3-ol using DVB/CAR/PDMS fibers. It is important to note that these studies investigated different types and concentrations of CD, materials of SPME fiber, and base materials of soymilk. As a result, the types and contents of the identified beany flavors were slightly different from those of the current study. However, the decreases in overall compounds pertaining to beany flavor were similarly induced by the addition of CD. In this study, based on GC–MS, the optimal conditions for efficient decreases in beany flavor compounds, including 2-pentylfuran, was the –CD-4 sample.

### 3.2. PCA of Volatile Components by GC–MS

PCA was used to gain a comprehensive understanding of the differences in volatile compounds among the ISP suspensions with/without CD (Figure 1). Alkanes, alkenes, alcohols, ketones, aldehydes, and furan were absorbed by the SPME fiber from the samples. Figure 1 shows that the two most important principal components (PC1 and PC2) extracted by PCA were responsible for 53.71% and 36.87% of the total variance, respectively. +CD-0 and −CD-0 located in the positive area of both PC1 and PC2 and were related to higher contents of alcohols, ketones, and aldehydes, which contained compounds of beany flavor, such as 1-octen-3-ol, benzaldehyde, hexanal, and 2-heptanones. Furthermore, +CD 1-4 were positioned in the higher left quadrant, whereas −CD 1-4 were positioned in the lower left quadrant. The positions of the +CD 1-4 samples were related to alkanes, alkenes and furans originating from CD. In particular, 2-pentylfuran was associated with the characteristic beany flavor. Therefore, PC2 illustrates the differences between the presence (+) and absence (−) of flavor-entrapped CD. The absence (−) of flavor compounds in the −CD group can be attributed to the removal of flavor-entrapped CD.

### 3.3. Sensory Evaluation of ISP Colloids and Yuba Films

A sensory evaluation of the flavor strength, beany flavor, and sweetness of ISP-based yuba films fabricated with different CD concentrations is shown in Table 2. +CD-1 and −CD-1 were not significantly different from +CD-0 and −CD-0 for all three parameters. The flavor strength and beany flavor of +CD-2 and −4 and –CD-2 and −4 were notably decreased with increasing concentrations. Sweetness was perceived to be less strong in all samples, as seen in another study on cooked oatmeal [19]. However, sweetness was significantly reduced for the −CD-4 samples. In the comparisons between the +CD and −CD groups, significant sensorial recognition was observed with 4% CD addition.

Based on the GC–MS results, alkanes and alkenes, which were at higher intensities for the +CD group, may have had little contribution to flavor due to the large threshold values of these compounds in flavor perception [22]. On the other hand, 2-heptanone, hexanal, and 1-octen-3-ol, with low perception thresholds, were related to flavor profiles such as penetrating fruity odor and mushroom-like [23]. It is likely that the decreases in these compounds induced by the addition of CD might have influenced the sensory differentiation of beany flavor. As shown in the GC–MS profile results, for +CD-2 and −4, most related compounds were significantly reduced with the exception of 2-pentylfuran, which is related to the characteristic beany flavor. The sensory perception of a mixture of various volatile compounds might be different from that of individual compounds.

According to other studies on the sensory evaluation of soymilk, sensory differences were perceived with the addition of 0.75% β-CD in soymilk during heat treatment [21], while the addition of 0.5% α-CD or γ-CD did not cause any significant difference [20]. In this study, sensory properties up to 1% CD remained unaffected, regardless of the presence or absence of excessive flavor-entrapped CD. However, a higher concentration of CD (2 or 4%) and the absence (−) of flavor-entrapped CD in ISP colloids could be useful in the reduction of sensorial perception of beany flavor.

### 3.4. Color Properties and Puncture Strength of Yuba Film Based on ISP with/without CD

The color properties of the yuba films affected by the +CD and −CD groups are shown in Table 3. The lightness (L) values of the films showed increasing trends with increasing CD concentration for both groups. The +CD-0 and −CD-0 films showed glossy, transparent, and brown color characteristics, and differences between 0 and 1% CD were imperceptible. Noticeable differences between +CD-4 and −CD-4 were observed with lightness (L) and redness (a) values. However, the addition of CD higher than 2% increased the opacity and brightness but decreased the brownness of yuba films. The color parameters of −CD-4 were similar to those of −CD-2. This result indicate that the absence of CD complexes did not significantly influence the color properties of the films.

The puncture strength and deformation of yuba films made using ISP colloids of the +CD and –CD groups are also shown in Table 2. With increasing CD concentration, puncture strength and deformation were found to decrease. Films with the highest concentration (+CD-4) had the lowest puncture strength and deformation. However, in the −CD group, puncture and deformation strength were not significantly different within the different treatments. The presence or absence of CD did not influence the puncture strength of the films until 1% CD but decreased for +CD-2 and 4 compared to –CD-2 and 4. A similar result was observed in another study, wherein a high carbohydrate concentration lowered the strength of yuba film [2]. In this study, CD, a carbohydrate derivative, could have played a similar role in decreasing the strength of yuba films. Thus, the addition of 1% CD could result in textural properties similar to those of films without CD.

### 3.5. SEM Observation of Yuba Film Based on ISP with/without CD

SEM images of transverse sections of yuba films with thicknesses of ~200 µm are shown in Figure 2. The film without CD showed a smooth surface and uniform matrix in the transverse section. With the addition of CD, granular structures appeared to have dispersed throughout the matrices, indicating the presence of specific complexes in the yuba films. In the −CD group, increases in complexes were not observed as the concentrations of CD increased due to the removal of the excessive flavor-entrapped CD. In the samples below 2% concentration, the CD complexes were incorporated from beneath films (the lower surface), while the upper surface of film formation showed a smooth characteristic. In the +CD group, 4% CD addition caused complexes to accumulate on both the upper and lower surfaces. The accumulation of CD on the surfaces of the yuba film decreased the surface glossiness due to the formation of prominent complexes. It is likely that the presence of excessive CD complexes could have decreased the protein interactions within the films, in turn influencing its texture.

## 4. Conclusions

The optimal concentration of CD was investigated for the improvement of the flavor and texture of yuba film. CD addition contributed to the significant decreases in beany-flavor-related compounds, such as 1-octen-3-ol, 2-heptanone, benzaldehyde, and hexanal in ISP colloids, which were used for yuba film formation. CD complexes on the lower surface of yuba film decreased the glossiness, which is considered to negatively impact the organoleptic quality of meat substitutes. The application of a higher concentration of CD (2 or 4%) and the removal of excessive flavor-entrapped CD could be beneficial in the reduction of the sensorial perception of beany flavor. Although the strength of films remained unaffected up to 1% CD, it was affected when the concentration was raised to 2 or 4% CD, thereby influencing the textural properties of the yuba films. The absence of excessive flavor-entrapped CD (−CD) was not significantly affected to the reduction of the puncture strength and deformation values. Thus, yuba films with improved color values, textural properties and lower beany flavors have the potential to be utilized as meat analogs.

## Figures and Tables

**Figure 1 foods-09-00818-f001:**
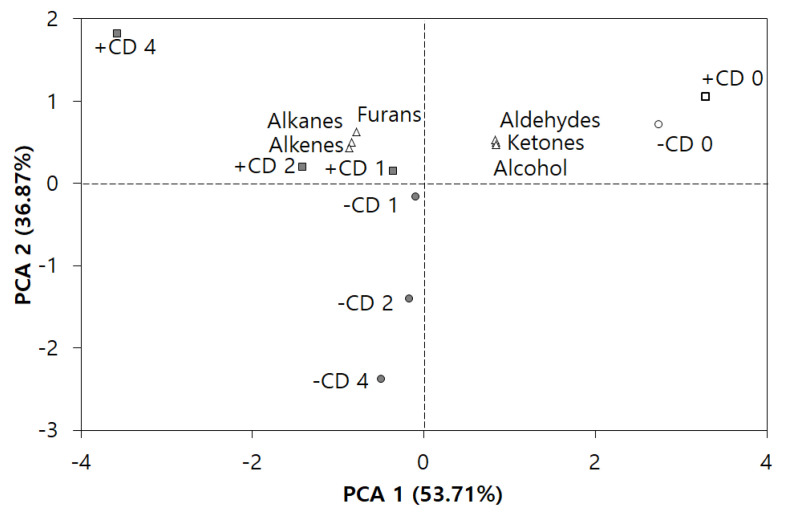
Principal component analysis of the volatile compound natures affected by CD concentrations (0, 1, 2, and 4) and the presence (+) or absence (−) of excessive flavor-entrapped CD (+CD and −CD groups) in ISP colloid.

**Figure 2 foods-09-00818-f002:**
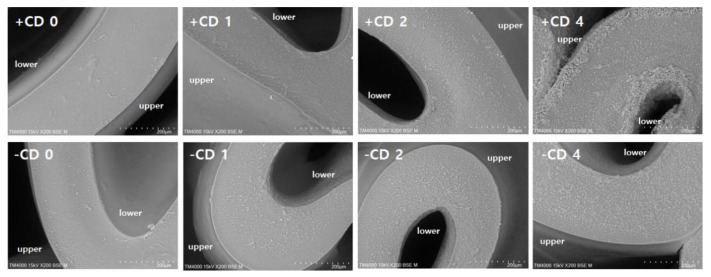
Scanning electron microscopy (SEM) images of transverse sections of ISP-based yuba films depending on different CD concentrations (0, 1, 2, and 4) and the presence (+) or absence (−) of excessive flavor-entrapped CD (+CD and −CD groups).

**Table 1 foods-09-00818-t001:** Gas chromatography–mass spectrometry (GC–MS) results of volatile components as affected by cyclodextrin in isolated soy protein (ISP) colloids with/without centrifuging ^(1).^

RT^(2)^ RI^(3)^		Compound Natures Data File Name	+CD Group ^(4)^				−CD Group			
			0 ^(5)^	1	2	4	0	1	2	4
		**Alkanes**								
8.39	800>	2,4-Dimethylheptane	0.1 ^dE (6)^	1.4 ^cC^	1.9 ^bB^	4.7 ^aA^	0.1 ^bE^	0.9 ^aD^	0.9 ^aD^	0.8 ^aD^
14.58	1047	2,5,6-Trimethyldecane	0.0 ^dF^	0.6^cC^	0.9 ^bB^	2.6 ^aA^	0.0 ^cF^	0.4 ^aD^	0.2 ^bDE^	0.2 ^bE^
5.95	800>	2-Methyl-2-methoxypropane	0.2 ^(7)^	0.1	0	0	0.1	0.1	0.1	0
7.26	800>	2-Methylheptane	0.0 ^dE^	0.1 ^cC^	0.2 ^bB^	0.5 ^aA^	0.0 ^bDE^	0.1 ^aCD^	0.1 ^aC^	0.1 ^aC^
14.75	1051	4-Methyldecane	0.0 ^dF^	0.6 ^cC^	1.1 ^bB^	3.1 ^aA^	0.1 ^cF^	0.4 ^aCD^	0.3 ^abDE^	0.2 ^bcEF^
7.38	800>	4-Methylheptane	0.0 ^cD^	0.1 ^cCD^	0.2 ^bB^	0.5 ^aA^	0.0 ^cD^	0.2 ^abBC^	0.1 ^bBCD^	0.2 ^aB^
9.7	819	4-Methyloctane	0.1 ^dD^	0.5 ^cC^	0.8 ^bB^	2.1 ^aA^	0.1 ^bD^	0.4 ^aC^	0.4 ^aC^	0.4 ^aC^
16.24	1093	5-Methyldecane	0.0	0.1	0.2	0.6	0.0	0.1	0.1	0.1
14.34	1040	Decane	0.3 ^dD^	1.8 ^cC^	3.9 ^bB^	10.6 ^aA^	0.5 ^bD^	1.6 ^aC^	1.4 ^aC^	1.6 ^aC^
21.89	1244	Dodecane	0.1 ^dD^	0.8 ^cC^	2.4 ^bB^	8.9 ^aA^	0.1 ^bD^	0.8 ^aC^	0.7 ^aC^	0.9 ^aC^
15.88	1083	Tetradecane	0.0 ^dE^	0.7 ^cC^	1.0 ^bB^	2.6 ^aA^	0.0 ^cF^	0.4 ^aD^	0.2 ^bDE^	0.2 ^bcDF^
23.51	1288	Hexadecane	0.0 ^dE^	0.8 ^cC^	1.5 ^bB^	5.4 ^aA^	0.0 ^cE^	0.4 ^aD^	0.2 ^abDE^	0.2 ^bcDE^
8.13	800>	Octane	0.0 ^dE^	0.3 ^cCD^	0.5 ^bB^	1.2 ^aA^	0.1 ^cE^	0.2 ^bD^	0.3 ^abCD^	0.4 ^aBC^
15.68	1077	Tridecane	0.0 ^dF^	1.5 ^cC^	2.2 ^bB^	6.5 ^aA^	0.1 ^cF^	0.7 ^aD^	0.4 ^bDE^	0.3 ^bcDE^
17.68	1131	Undecane	0.0 ^dF^	0.6 ^cC^	1.1 ^bB^	3.4 ^aA^	0.0 ^cF^	0.4 ^aD^	0.2 ^bE^	0.2 ^bE^
		**Total alkanes**	**0.8 ^dE^**	**10.0 ^cC^**	**17.9 ^bB^**	**52.7 ^aA^**	**1.2 ^bE^**	**7.1 ^aD^**	**5.6 ^aD^**	**5.8 ^aD^**
		**Alkenes**								
27.5	1401	1,2-Dimethyl-4-ethylbenzene	0.2 ^cE^	0.6 ^cC^	1.1 ^bB^	1.7 ^aA^	0.2 ^bE^	0.6 ^aC^	0.5 ^aCD^	0.4 ^abCD^
26.41	1370	1,3,5-Trimethylbenzene	0.2 ^cC^	0.4 ^bcC^	0.6 ^abB^	0.7 ^aA^	0.2 ^abC^	0.3 ^aBC^	0.2 ^bC^	0.1 ^bC^
29.62	1464	1,3-Di-tert-butylbenzene	1.5 ^dD^	8.5 ^aA^	4.6 ^bB^	3.0 ^cC^	1.5 ^cD^	8.1 ^aA^	3.5 ^bC^	1.2 ^cD^
10.18	831	2,4-Dimethyl-1-heptene	0.1 ^bE^	0.3 ^bDE^	0.6 ^aAB^	0.8 ^aA^	0.1 ^cE^	0.5 ^abBCD^	0.4 ^bCD^	0.6 ^aABC^
21.05	1222	dl-Limonene	0.0 ^cC^	1.7 ^cC^	5.7 ^bB^	12.8 ^aA^	0.0 ^bC^	1.5 ^aC^	1.7 ^aC^	1.6 ^aC^
15.06	1060	Methylbenzene	0.2 ^bBC^	0.2 ^bBC^	0.4 ^abB^	0.6 ^aA^	0.1 ^bC^	0.2 ^abBC^	0.2 ^abBC^	0.3 ^aBC^
		**Total alkenes**	**2.2 ^cD^**	**11.7 ^bB^**	**13.0 ^bB^**	**19.6 ^aA^**	**2.1 ^dD^**	**11.2 ^aB^**	**6.5 ^bC^**	**4.2 ^cCD^**
		**Alcohols**								
29.46	1459	1-Octen-3-ol	2.9 ^aB^	1.4 ^bC^	1.1 ^cD^	0.9 ^cDE^	3.6 ^aA^	1.5 ^bC^	1.1 ^cD^	0.6 ^dE^
22.45	1260	1-Pentanol	0.5 ^aB^	0.2 ^bC^	0.1 ^cDE^	0.1 ^cE^	0.7 ^aA^	0.2 ^bCD^	0.1 ^bCDE^	0.1 ^bDE^
30.92	1502	2-Ethyl-1-hexanol	1.4 ^aA^	0.4 ^cB^	0.4 ^cB^	0.3 ^cB^	1.5 ^aA^	0.5 ^bB^	0.4 ^bB^	0.2 ^cC^
		**Total alcohols**	**4.8 ^aA^**	**2.0 ^bB^**	**1.6 ^cBC^**	**1.3 ^cC^**	**5.8 ^aA^**	**2.2 ^bB^**	**1.6 ^bcBC^**	**0.9 ^cC^**
		**Aldehydes**								
26.17	1363	2-Ethylhexenal	0.7 ^aB^	0.5 ^bCD^	0.5 ^bcCDE^	0.4 ^cDEF^	0.9 ^aA^	0.6 ^bC^	0.3 ^cEF^	0.2 ^cF^
39.14	1676	4-Methylbenzaldehyde	4.4 ^aA^	0.6 ^bCD^	0.5 ^bCD^	0.6 ^bCD^	1.5 ^aB^	1.3 ^aBC^	0.6 ^bCD^	0.3 ^bD^
32.86	1549	Benzaldehyde	3.8 ^aA^	2.2 ^bBC^	1.6 ^cCD^	1.3 ^cDE^	4.1 ^aA^	2.4 ^bB^	1.5 ^cD^	0.9 ^cE^
16.65	1104	Hexanal	10.0 ^aA^	7.8 ^bB^	6.5 ^bcBC^	5.7 ^cC^	10.0 ^aA^	8.0 ^bB^	6.8 ^bBC^	6.5 ^bBC^
40.05	1692	p-Propylbenzaldehyde	1.0 ^aA^	0.8 ^bB^	0.6 ^cCD^	0.4 ^dE^	1.0 ^aA^	0.7 ^bC^	0.5 ^cDE^	0.2 ^dF^
		**Total aldehydes**	**19.9 ^aA^**	**11.9 ^bBC^**	**9.7 ^bcCD^**	**8.4 ^cD^**	**17.5 ^aA^**	**13.0 ^bB^**	**9.7 ^cCD^**	**8.1 ^cD^**
		**Ketones**								
38.32	1661	1-(3-Butyl-2-oxiranyl)ethanone	1.7 ^aA^	0.7 ^bB^	0.5 ^bBC^	0.2 ^cD^	1.6 ^aA^	0.6 ^bB^	0.4 ^cCD^	0.2 ^cD^
20.44	1205	2-Heptanone	13.8 ^aA^	10.9 ^abB^	8.0 ^bcC^	5.2 ^cD^	15.9 ^aA^	9.1 ^bBC^	7.2 ^cCD^	5.2 ^dD^
28.04	1417	2-Nonanone	0.2 ^BC^	0.4 ^A^	0.4 ^A^	0.3 ^AB^	0.3 ^aAB^	0.4 ^aA^	0.2 ^bBC^	0.1 ^cC^
34.85	1597	3,5-Octadien-2-one	0.9 ^aB^	0.8 ^aBC^	0.7 ^abBCD^	0.5 ^bCD^	1.3 ^aA^	0.6 ^bCD^	0.6 ^bCD^	0.4 ^bD^
28.65	1435	3-Octen-2-one	1.1	1.3	1.1	0.9	1.1	1	0.9	0.7
		**Total ketones**	**17.7 ^aA^**	**14.1 ^abB^**	**10.7 ^bcC^**	**7.1 ^cD^**	**20.2 ^aA^**	**11.7 ^bBC^**	**9.3 ^cCD^**	**6.6 ^dD^**
		**Furan**								
22.27	1255	2-Pentylfuran	6.2 ^cD^	15.0 ^bC^	20.9 ^abB^	22.4 ^aA^	7.4 ^bD^	13.2 ^aC^	7.1 ^bD^	4.7 ^cD^
		**Total furans**	**6.2 ^cD^**	**15.0 ^bC^**	**20.9 ^abB^**	**22.4 ^aA^**	**7.4 ^bD^**	**13.2 ^aC^**	**7.1 ^bD^**	**4.7 ^cD^**

^(1)^ Identification of volatile compounds was carried out by comparing the MS spectrum. In the comparison of MS spectrum, SI should be more than 800. The results were obtained based on the relative peak ratio of each compound based on the internal standard. ^(2)^ RT means the retention time (min) of each compound. ^(3)^ RI was calculated in relation to the retention time of n-alkane series. ^(4)^ The +CD and –CD groups indicate the presence (+) or absence (−) of excessive flavor-entrapped CD, respectively. ^(5)^ Numbers 0-4 represent the concentrations of CD (% (*w*/*w*)) in ISP colloids. ^(^^6^^)^ Data with different capital and small letters of superscript are significantly different (*p* < 0.05) within the same line and group, respectively. ^(7)^ Data without superscript letters are not significantly different (*p* > 0.05) among the samples.

**Table 2 foods-09-00818-t002:** Sensory evaluation of flavor strength, beany flavor, and sweetness in ISP colloids depending on the CD concentrations and centrifugation process ^(1).^

Group ^(2)^	CD con.	Parameters		
		Flavor Strength	Beany Flavor	Sweetness
+CD	0	5.0 ± 0.0 ^A^^a (3)^	5.0 ± 0.0 ^a^^AB^	5.0 ± 0.0 ^a^^A^
	1	4.9 ± 0.7 ^a^^bAB^	4.9 ± 0.6 ^a^^AB^	4.9 ± 0.2 ^a^^A^
	2	4.3 ± 0.8 ^ab^^ABC^	4.5 ± 0.6 ^a^^ABC^	4.7 ± 0.4 ^a^^AB^
	4	3.8 ± 1.4 ^b^^C^	3.9 ± 0.5 ^b^^C^	4.4 ± 0.5 ^b^^B^
−CD	0	5.3 ± 0.4 ^a^^A^	5.5 ± 0.5 ^a^^A^	4.9 ± 0.2 ^a^^A^
	1	4.7 ± 0.5 ^a^^bABC^	4.7 ± 0.4 ^a^^AB^	4.8 ± 0.4 ^a^^AB^
	2	4.1 ± 0.7 ^b^^BC^	4.3 ± 0.6 ^b^^BC^	4.4 ± 0.5 ^b^^B^
	4	2.9 ± 0.7 ^c^^D^	3.1 ± 0.7 ^c^^D^	3.9 ± 0.5 ^c^^C^

^(1)^ Data are means ±standard deviations (n = 10). The score of +CD was 5 as a standard for panelists. ^(2)^ Here, +CD and −CD indicate the presence or absence of excessive flavor-entrapped CD, respectively. Numbers 0-4 represent the concentrations of CD (% (*w*/*w*)) in ISP colloids. ^(3)^ Data with different capital and small letters of superscript are significantly different (*p* < 0.05) within the same line and group, respectively.

**Table 3 foods-09-00818-t003:** Color values and puncture strength properties of yuba films based on different β-cyclodextrin (CD) concentrations ^(1).^

Group ^(2)^	CD con.	Parameters				
		L-value	a-value	b-value	Puncture Strength (N)	Puncture Deformation (mm)
+CD	0	50.8 ± 1.67 ^c^^C (3)^	2.03 ± 0.54 ^cC^	18.9 ± 1.38 ^bC^	3.08 ± 0.33 ^a^^A^	7.19 ± 1.15 ^a^^A^
	1	51.9 ± 0.73 ^c^^C^	3.00 ± 0.59 ^b^^B^	19.9 ± 1.39 ^b^^BC^	2.71 ± 0.49 ^a^^A^	6.51 ± 1.94 ^a^^A^
	2	62.7 ± 0.88 ^b^^B^	4.13 ± 0.57 ^aA^	25.2 ± 0.93 ^aA^	1.62 ± 0.23 ^b^^B^	3.23 ± 0.87 ^b^^B^
	4	71.8 ± 1.77 ^a^^A^	4.10 ± 0.22 ^aA^	24.4 ± 0.67 ^aA^	1.12 ± 0.22 ^bB^	1.23 ± 0.18 ^c^^B^
−CD	0	50.5 ± 2.30 ^b^^C^	1.75 ± 0.13 ^cC^	18.9 ± 0.47 ^cC^	2.87 ± 0.60 ^a^^A^	7.15 ± 2.12 ^a^^A^
	1	51.8 ± 1.69 ^b^^C^	2.98 ± 0.28 ^bB^	21.0 ± 0.40 ^bB^	3.07 ± 0.25 ^a^^A^	6.99 ± 3.49 ^a^^A^
	2	60.4 ± 0.80 ^a^^B^	4.23 ± 0.33 ^a^^A^	25.3 ± 1.24 ^a^^A^	2.83 ± 0.44 ^a^^A^	6.37 ± 2.03 ^a^^A^
	4	61.4 ± 2.05 ^a^^B^	4.15 ± 0.25 ^a^^A^	25.4 ± 0.26 ^a^^A^	2.84 ± 0.44 ^a^^A^	6.21 ± 2.04 ^a^^A^

^(1)^ Data are means ±standard deviations (n = 5 of each triplicate determination). ^(2)^ Here, +CD and −CD indicate the presence or absence of excessive flavor-entrapped CD, respectively. Numbers 0–4 represent the concentrations of CD (% (*w*/*w*)) in ISP colloids. ^(3)^ Data with different capital and small letters of superscript are significantly different (*p* < 0.05) within the same line and group, respectively.

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
