# Peer review of "β-Cyclodextrin-Mediated Beany Flavor Masking and Textural Modification of an Isolated Soy Protein-Based Yuba Film"

_foods, 2020, doi:10.3390/foods9060818_

Round 1
Reviewer 1 Report
General comments: The authors present a very interesting research about β-Cyclodextrin-Mediated Beany Flavor Masking and Textural Modification of an Isolated Soy Protein-Based Yuba Film.
This work is well structed, presented and written. The all paper is well described, and it has a strong impact on research. Each part of manuscript is carefully described. The results are consistent. The soundness and the novelty of the work is remarkable.
However, there are small curiosities. Could the author explain these range (ISP (5%, w/w) and CD (0, 1, 2, and 4%, w/w)) of ISP and CD? Do they not think it is too high as a ratio?
In table 1 which reports, the GC-MS results of volatile components as affected by cyclodextrin in ISP colloids with/without centrifuging, are reported twice the same molecules 4-Methyldecane labelled with 2, what the author mean?
Author Response
The authors present a very interesting research about β-Cyclodextrin-Mediated Beany Flavor Masking and Textural Modification of an Isolated Soy Protein-Based Yuba Film.
Point 1: This work is well structed, presented and written. The all paper is well described, and it has a strong impact on research. Each part of manuscript is carefully described. The results are consistent. The soundness and the novelty of the work is remarkable.
However, there are small curiosities. Could the author explain these range (ISP (5%, w/w) and CD (0, 1, 2, and 4%, w/w)) of ISP and CD? Do they not think it is too high as a ratio?
Response 1: We appreciate the time and efforts by reviewing this manuscript. In this study, ISP 5% colloid was used based on soymilk contained about 5% protein for traditional yuba formation. As reviewer commented, we also agree that 4% CD was too high because CD has water solubility of below 2% and when CD ratio compared with ISP concentrations. In this study, our question was effects of removing flavor-entrapped CD, therefore excess CD was applied.
Point 2: In table 1 which reports, the GC-MS results of volatile components as affected by cyclodextrin in ISP colloids with/without centrifuging, are reported twice the same molecules 4-Methyldecane labelled with 2, what the author mean?
Response 2: Some compounds detected in separated peaks in GC-MS profiles were identified same compound. In that case, we did not sum up the values and let them separate.
Reviewer 2 Report
Comments:
Line 33 - Please check carefully if in the text citations have been prepared in accordance with the instructions for the authors (…it should be done numbering again).
Lines 66-72 – it should be specified the nomenclature for 0-4 in this paragraph (es. Numbers 0-4 represent the concentrations of CD (w/w, %) in ISP colloids. So the legend of the tables and figures could be more synthetic.
Line 98 – please delete a bracket.
Line 96-110; Line 229: two paragraphs 2.5 and 2.6 could be united in a single paragraph entitles “Physical analysis “
Please include in “Material and Methods” a section of procedure for SEM observation.
Line 117. It should not 1st …. (this sentence should be rewritten, I don’t understand if sample name C-0 has same meaning of C-0 in other tables or figures.
Statistical analysis results are not shown very clearly in the tables. A suggestion would be analyse significant differences using ANOVA and post hoc procedure for both group (-CD, +CD) and concentration of CD 0-4 %; it could be used different small and capital superscript letters, respectively.
Was there a correlation among sensory characteristics and volatile profile detected by GC-MS?
Author Response
Comments:
Line 33 - Please check carefully if in the text citations have been prepared in accordance with the instructions for the authors (…it should be done numbering again).
- We thank the reviewer for the recommendation. As reviewer`s suggestion, we checked and corrected the text citations.
Lines 66-72 – it should be specified the nomenclature for 0-4 in this paragraph (es. Numbers 0-4 represent the concentrations of CD (w/w, %) in ISP colloids. So the legend of the tables and figures could be more synthetic.
- We added the nomenclature for 0-4 in the line 72.
Line 98 – please delete a bracket.
Line 96-110; Line 229: two paragraphs 2.5 and 2.6 could be united in a single paragraph entitles “Physical analysis “
- We followed reviewer`s comment.
Please include in “Material and Methods” a section of procedure for SEM observation.
- We added a section for SEM observation.
Line 117. It should not 1st …. (this sentence should be rewritten, I don’t understand if sample name C-0 has same meaning of C-0 in other tables or figures.
- The CD-0 was provided at first to panelists as the standard in comparison with other samples.
- The panelists evaluated CD-0 as the 1st sample, and then randomly served the other samples with offering a 3 min rest between individual samples.
Statistical analysis results are not shown very clearly in the tables. A suggestion would be analyse significant differences using ANOVA and post hoc procedure for both group (-CD, +CD) and concentration of CD 0-4 %; it could be used different small and capital superscript letters, respectively.
- Statistic analysis results inserted on table 1 and indicated the statistic results using small and capital superscript letters as the reviewer`s suggestion.
Was there a correlation among sensory characteristics and volatile profile detected by GC-MS?
- We have not considered a correlation among sensory characteristics and volatile profiles. It should be a useful study for estimating GC-MS profile. It is required further research.
Reviewer 3 Report
In this paper β-cyclodextrin (CD) was incorporated as potential ingredient for the improvement of flavor and physical properties of yuba films. The study seems original. The experiments are well designed. The conclusions were confirmed by the obtained results. Please consider the following comments to revise the current version of the manuscript:
- Page 1-line 35/48: please correct the reference style according to the journal rules “(Zhang et al., 2017[23]). », (Faridi Esfanjani & Jafari, 2016 [24])…
- The ANOVA comparison Is missed in table one.
- From figure1 please indicate if there is a particular classification of different samples (CD concentration and excessive flavor-entrapment) based on PCA analysis.
- Are there any results on sensory study done based on the obtained yuba films, if yes please include it to confirm the current finding?
- Conflicts of Interest: “This authors” correct to “The authors”
Author Response
- Page 1-line 35/48: please correct the reference style according to the journal rules “(Zhang et al., 2017[23]). », (Faridi Esfanjani & Jafari, 2016 [24])…
- We thank the reviewer for the recommendation. We checked and corrected the reference style.
- The ANOVA comparison Is missed in table one.
- The ANOVA comparison each compound was performed and inserted in table 1.
- From figure1 please indicate if there is a particular classification of different samples (CD concentration and excessive flavor-entrapment) based on PCA analysis.
- Statistical analysis of each compound shown in table 1 was performed and inserted with superscript letters.
- Are there any results on sensory study done based on the obtained yuba films, if yes please include it to confirm the current finding?
- We did not evaluate results of sensory study based on the yuba films.
- Conflicts of Interest: “This authors” correct to “The authors”
- Corrected
Reviewer 4 Report
This study lacks a piece of important information. There is no sufficient evidence that CDs were forming complexes with undesirable flavor compounds.
How can the authors convince the readers that they have complete complex formation? Flavors should be able to fit the CD cavities to form the complexes. These flavor compounds have various molecular sizes. Sensory tests do not provide enough evidence.
Author Response
Point 1: This study lacks a piece of important information. There is no sufficient evidence that CDs were forming complexes with undesirable flavor compounds.
How can the authors convince the readers that they have complete complex formation?
Response 1: Main objective of this study was to remove unattractive flavors from soy by CD. The authors believe that GC data would be an evidence to form a complex, since part of volatile compounds were removed with CD concentration.
Point 2: Flavors should be able to fit the CD cavities to form the complexes. These flavor compounds have various molecular sizes. Sensory tests do not provide enough evidence.
Response 2: The authors agree to reviewer’s comment that sensory test provided just part of characteristics of CD-treated Yuba. However, our sample is not commercial products (type of soy-based meat analog) but a beginning platform of the meat analog. This study was focused on flavor masking, particularly masking beany flavor which is regarded as the most problem in soy-based meat analog.